# Integrating Patient-Based Real-Time Quality Control (PBRTQC) in a New Field: Inter-Comparison between Biochemical Instrumentations with LDL-C

**DOI:** 10.3390/diagnostics14090872

**Published:** 2024-04-23

**Authors:** Jingyuan Wang, Chedong Zhao, Linlin Fan, Xiaoqin Wang

**Affiliations:** The First Affiliated Hospital of Xi’an Jiaotong University, Xi’an 710061, China; wangjingyuan@xjtu.edu.cn (J.W.); zkg0212@xjtufh.edu.cn (C.Z.); fviolinlin@xjtufh.edu.cn (L.F.)

**Keywords:** PBRTQC, EWMA, quality control, biochemical analyzer

## Abstract

Background: Patient-based real-time quality control (PBRTQC) can be a valuable tool in clinical laboratories due to its cost-effectiveness and constant monitoring. More focus is placed on discovering and improving algorithms that compliment conventional internal control techniques. The practical implementation of PBRTQC with a biochemical instrument comparison is lacking. We aim to evaluate PBRTQC’s efficacy and practicality by comparing low-density lipoprotein cholesterol (LDL-C) test results to ensure consistent real-time monitoring across biochemical instrumentations in clinical laboratories. Method: From 1 September 2021 to 30 August 2022, the First Affiliated Hospital of Xi’an Jiaotong University collected data from 158,259 both healthy and diseased patients, including 84,187 male and 74,072 female patients, and examined their LDL-C results. This dataset encompassed a group comprising 50,556 individuals undergoing health examinations, a group comprising 42,472 inpatients (IP), and a group comprising 75,490 outpatients (OP) for the PBRTQC intelligent monitoring platform to conduct daily tests, parameter configuration, program development, real-time execution, and performance validation of the patients’ data. Moreover 40 patients’ LDL-C levels were assessed using two biochemical analyzers, designated as the reference and comparator instruments. A total of 160 LDL-C results were obtained from 40 both healthy and diseased patients, including 14 OP, 16 IP, and 10 health examination attendees, who were selected to represent LDL-C levels broadly. Two biochemical instruments measured LDL-C measurements from the same individuals to investigate consistency and reproducibility across patient statuses and settings. We employed exponentially weighted moving average (EWMA) and moving median (MM) methods to calculate inter-instrument bias and ensure analytical accuracy. Inter-instrument bias for LDL-C measurements was determined by analyzing fresh serum samples, different concentrations of quality control (QC), and commercialized calibrators, employing both EWMA and MM within two assay systems. The assessment of inter-instrumental bias with five different methods adhered to the external quality assessment standards of the Clinical Laboratory Center of the Health Planning Commission, which mandates a bias within ±15.0%. Result: We calculated inter-instrument comparison bias with each of the five methods based on patient big data. The comparison of fresh serum samples, different concentrations of QC, commercialized calibrators, and EWMA were all in the permissive range, except for MM. MM showed that the bias between two biochemical instruments in the concentration ranges of 1.5 mmoL/L–6.2 mmoL/L exceeded the permissible range. This was mainly due to the small number of specimens, affected by variations among individual patients, leading to increased false alarms and reduced effectiveness in monitoring the consistency of the inter-instrumental results. Moreover, the inter-comparison bias derived from EWMA was less than 3.01%, meeting the 15% range assessment criteria. The bias result for MM was lower than 24.66%, which was much higher than EWMA. Thus, EWMA is better than MM for monitoring inter-instrument comparability. PBRTQC can complement the use of inter-comparison bias between biochemical analyzers. EWMA has comparable inter-instrument comparability monitoring efficacy. Conclusions: The utilization of AI-based PBRTQC enables the automated real-time comparison of test results across different biochemical instruments, leading to a reduction in laboratory operating costs, enhanced work efficiency, and improved QC. This advanced technology facilitates seamless data integration and analysis, ultimately contributing to a more streamlined and efficient laboratory workflow in the biomedical field.

## 1. Introduction

LDL-C has long been established as a critical marker for assessing cardiovascular disease (CVD) risk. Its concentration in serum is considered the dominant clinical parameter for evaluating a patient’s predisposition to develop CVD, a leading cause of morbidity and mortality worldwide [1]. In the realm of disease screening, monitoring, and diagnosis, biochemical instrumentation is fundamental to laboratory procedures in analyzing biochemical markers to produce results that provide physicians with critical information for effective disease management. This underscores the growing reliance on biochemical instrumentation in laboratories to support healthcare professionals in delivering optimal patient care. Thus, an inter-comparison between instrumentations for quality assurance is essential for laboratory QC [2]. Integrating PBRTQC into routine laboratory operations for LDL-C measurements not only facilitates the accurate assessment of CVD risk but also supports a patient-centered approach by incorporating real-time QC measures.

The International Organization for Standardization (ISO) 15,189 standard stipulates that when laboratories use two or more sets of testing systems to test the same item, they should establish a program for comparing the results of patients’ samples within the clinically appropriate range to ensure the consistency of the test results. For inter-instrument comparisons, laboratories generally use periodic inter-instrumental comparisons of fresh venous blood samples, a protocol that is not capable of continuously monitoring inter-instrumental results on a day-to-day basis and thus may increase QC risk. Domestic laboratories have reported using multiple instruments to test the same venous blood samples every day for inter-instrument comparisons to ensure the consistency of results, which is easy to operate and has no matrix effect. However, this invariably increases the operating costs of the laboratory [3]. Another indirect application of the inter-comparison of various analyzers is traditional internal quality control (IQC). It has the disadvantages of higher costs, more labor, lower error detection, erroneous error, and higher false alarms [4].

PBRTQC is a QC method that monitors and evaluates the stability of the analytical performance of the testing process using real-time and continuous patient test results [5,6]. Essentially, more studies have found that PBRTQC has more advantages than IQC [5,7,8,9,10]. It can utilize specific datasets for the realistic stimulation of error detection, and it can become a complement for better quality assurance [11]. Hence, PBRTQC can be used to monitor instruments for improving the overall quality assurance for the laboratory. The advantages over IQC products include lower costs, no matrix effects, the ability to monitor continuously in real-time, and high sensitivity to pre-analytical errors [7,9].

PBRTQC consists of various computational procedures, such as the moving sum of outliers (MovSO) or positive patients, moving quartile (MQ), moving average (MA) [12], MM [5], and EWMA [13]. MM has high requirements for sample size. Wilson et al. explored that monitoring medians for data derived from approximately Gaussian distributions requires a sample size of 200 individual patient results [14], and if the amount of test data is too small, it will significantly impact data efficacy [4,15]. However, MM is more robust in handling skewed data and outliers, common in clinical laboratory measurements, ensuring that the analysis of inter-instrument bias is not distorted by these factors [5]. Also, a patient data-based QC method, EWMA, introduces weighting coefficients based on historical data that can adjust offset sensitivity for monitoring small offsets in the testing process, which is less affected by the sample size, and we attempted to use this algorithm in this study for the establishment of a real-time monitoring program for inter-instrumental comparisons based on patient data [5,16].

We planned to use a specialized intelligent software platform based on the International Federation for Clinical Chemistry and Laboratory Medicine (IFCC) proposal that meets the technical requirements and functional characteristics of PBRTQC. We chose EWMA, MM, and the daily fresh venous blood comparison method for consistency comparisons of the results of the two biochemistry analyzers in the laboratory. Our goal was to evaluate the value of PBRTQC for inter-instrument comparisons and to establish a more convenient, cost-effective, and efficient solution for monitoring the consistency of biochemistry analyzer results [17].

## 2. Methods

### 2.1. Data

In alignment with the legal requirements and the IFCC PBRTQC recommendations, all patient identifiers were meticulously removed to ensure data confidentiality and compliance. In accordance with established guidelines, it states that laboratories need to determine the comparison process, the equipment, and techniques to establish a method to ensure that the results of patient samples are comparable within the scope of clinical applicability [18]. 

### 2.2. Materials and Methodology

#### 2.2.1. Materials

##### Patient Data Collection

The PBRTQC intelligent monitoring platform was integrated into the local laboratory’s local server to automatically collect the serum LDL-C test results from all hospital healthy and diseased patients (158,259), including individuals getting health exams (50,556), IP (42,472), and OP (75,490) (84,187 male and 74,072 female) in the First Affiliated Hospital of Xi’an Jiaotong University from 1 September 2021 to 30 August 2022, which were then subjected to statistical normality testing, parameter setting, program construction, real-time execution, and performance confirmation. We specifically selected a cohort of 40 patients, comprising 25 males (ages ranging from 26 to 76 years) and 15 females (ages ranging from 37 to 68 years). This selection includes 2 healthy individuals and 38 patients with various conditions, thereby providing a broad representation of LDL-C concentrations for analysis. To ascertain the reliability of our data, samples from each patient were analyzed using both a reference instrument and a comparator biochemical analyzer. This dual-analysis approach ensured that the measured LDL-C levels were accurately reflected across the entire linear range of the methodology, consistent with the EP-09A2 guidelines.

##### Instruments and Reagents and Software Platform

The LST008AS-2 automatic biochemistry analyzer, which is manufactured by Hitachi with it’s headquarters in Tokyo, Japan, is the reference instrument, and the LST008AS-1 automatic biochemistry analyzer, which is also manufactured by Hitachi with its headquarters in Tokyo, Japan, is the comparator instrument. The LDL lipoprotein (lot number: 1151) assay reagent and calibrator (lot number: 1123) were manufactured by Hitachi. The multi-concentration level QC product (Lot No.: 26480) was from Burroughs, Inc. in Plymouth, MI, USA. Intelligent monitoring platform PBRTQC: an AI-driven, intelligent monitoring system for real-time QC of patient data (AI moving average intelligent monitoring platform, AI-MA), which works directly with Hitachi Intermediates for data collection, was developed by ShanghaiMorishu Medical Technology Co. in Shanghai, China.

##### Experimental Conditions

The laboratory operating instructions under the ISO15189 quality management system required IQC, calibration, and instrument maintenance for daily IQC results before testing project specimens [19]. Our biochemical instrumentation for the LDL-C measurement was firstly calibrated using NIST standards. Our laboratory then used commercially available control materials that closely resembled human blood sample matrices and analyte concentrations, specifically for the biomarker LDL-C. These materials were carefully selected and implemented according to manufacturer’s recommendations. To test the instruments’ precision and accuracy, control materials at low, normal, and high levels were analyzed daily before patient samples were analyzed. Each calibration and QC event were documented.

#### 2.2.2. Methodology

##### Establishment

The performance validation and selection of the optimal PBRTQC program was established based on AI-MA for parameter setting, the selection of quality objectives, QC rules, outlier rejection criteria, cut-off value range, weighting coefficients, and control limits. The QC performance of the EWMA method within the selected time range was also validated, and QC performance evaluation metrics such as accuracy and sensitivity were evaluated by selecting suitable parameters. The optimal PBRTQC method was selected for the real-time monitoring of LDL-C, and inter-instrument comparisons were performed for the analytical performance variation in each instrument. The EWMA algorithm calculation model is shown here:$${\bar{Z}}_{t+1}={\bar{Z}}_{t}+\lambda \varepsilon_{t}=\lambda Z_{t}+\left(1-\lambda \right){\bar{Z}}_{t}$$
where ${\bar{Z}}_{t+1}$ is the estimated value at point *t* + 1, ${\bar{Z}}_{t}$ is the estimated value at point *t*, $\varepsilon_{t}$ is the actual measured value at point *t*, and *λ* is a weighting factor, 0 < *λ* ≤ 1.

##### Comparison Program

Selection of comparison concentrations: Based on the sample selection’s concentration requirements of CLSI-EP9-A2 and the data distribution of LDL-C quantification in the laboratory, we analyzed the distribution characteristics of patients’ data using the AI-MA data platform. Four concentration ranges with a more centralized distribution of data, and those in the vicinity of the medical decision level, were selected as the truncation intervals used for the comparison.

Implementation of the comparison program: Following the EP-09A2 protocol, we took 38 healthy and 2 diseased patients (25 males (26–76 years) and 15 females (37–68 years)). Samples with concentration classes evenly distributed across the method’s linear range. Two rounds of testing were completed in five days of standard operation utilizing the reference apparatus (LST008AS-1) and the comparator apparatus (LST008AS-2), both of which are products of Hitachi, Ltd., headquartered in Tokyo, Japan, with eight samples performed on each day. The samples were performed either in the order of 1, 2, 3, 4, 5, 6, 7, 8 or 8, 7, 6, 5, 4, 3, 2, 1, resulting in 160 comparisons of LDL cholesterol fresh blood. The QC and calibration product results were determined between the two assay systems, respectively. (1) The 160 LDL-C fresh blood comparison data and the collected serum LDL test result data, from 7 November 2022 to 18 November 2022, were imported into the AI-MA intelligent monitoring platform for patient data QC, and data collected from 7 July 2021 to 1 June 2022 were used as the training set for EWMA.

Moreover, 160 fresh blood samples were collected from 40 participants, with 4 samples taken from each person on the same day to reduce biological variation and assure consistent assessments of LDL-C levels. Furthermore, 40 participants were subjected to a series of QC checks designed to minimize biological variation and ensure consistent assessments of LDL-C levels. These QC measures included the verification of sample integrity, the calibration of instruments prior to use, and the implementation of standardized procedures for LDL-C determinations. All sample collections and subsequent LDL-C determinations were conducted in the same laboratory. By using a single laboratory setting for all measurements, we calculated the cumulative mean and median of the EWMA quantitative results of 160 fresh specimens of LDL-C from the two instruments and the relative bias (Bias%) of each instrument from EWMA, MM, the calibrator method, and the fresh specimen comparison method. The final calculation was compared with the reference instrument to perform a comparison and comparability analysis between instruments with different concentration ranges.

In line with the requirements of EP9A2, 160 fresh specimen comparison data were compared with the patient data on the same day in the AI-MA platform in real time. The relative bias, standard deviation of difference (SD diff), and confidence intervals between the results of the reference instrument and the comparator instrument for the fresh specimen results were calculated. The bias of the patient data for EWMA and MM was calculated and compared with the bias results of the industry standards/allowable biological variation. We analyzed the bias results with the industry standard/biological variation allowable bias quality objectives for comparability between the three comparison methods of patient data, including EWMA, MM, and fresh specimens.

##### Statistical Analysis Methods

In line with the CLSI EP9A2 document, we statistically analyzed the bias of the new sample methods, including performing intra- and inter-method outlier exclusion, plotting linear regression analyses and bias hash plots, measuring regression formulas for the two methods, and measuring the bias and 95% confidence ranges at the level of medical judgment.
$$[{\hat{B}}_{c,low},{\hat{B}}_{c,high}]={\hat{B}}_{c}\pm 2S_{y\cdot x}\sqrt{\frac{1}{2N}+\frac{{{(X}_{c}-\bar{X})}^{2}}{\sum \sum {\left(X_{ij}-\bar{X}\right)}^{2}}}$$

Statistical method for methodological comparison bias of calibration and QC products: Relative bias of inter-instrument comparison—Bias% = (reference instrument result − comparator instrument result)/comparator instrument result × 100%.

Statistical method for methodological comparison bias of PBRTQC: Relative bias of inter-instrument comparison—Bias% = (mean value of EWMA of reference instruments − mean value of EWMA of comparator instruments)/mean value of EWMA of comparator instruments × 100%.

##### Judgment Criteria

The PBRTQC procedure was evaluated for optimal performance as evidenced by the ability to match warning data to quality events using the EWMA Z-score QC plot; the relatively stable and normally distributed distribution of patient data results and its performance met the criteria of an error detection probability of more than 90% and a false alarm rate of less than 5%.

The standard acceptance of the device was based on results derived from the CLSI EP9A2 fresh blood serum comparison method. If the degree of bias at the medical decision level was less than or equal to the quality needs of the laboratory, then the comparison was acceptable. We compared fresh blood serum with different methods, such as CLSI EP9A2, QC, calibrators, EWMA, and MM with inter-device bias using the NHS Clinical Laboratory Quality Assessment inter-laboratory requirements (≤±15.0%) as the decision criteria.

Consistency evaluation of inter-method comparison: The inter-instrument bias obtained by the two methods was compared with the required and permissible range, respectively, and if they both passed and the trend was the same, the two methods were equally effective in judging the consistency of inter-instrument results.

Figure 1 illustrates the comprehensive workflow of data collection and analysis employed in our study.

## 3. Results

### Comparison

The comparison of the results from the patient data for inter-instrument comparison between EWMA, MM, and the daily fresh blood comparison method, including IQC, calibrator, and fresh blood comparisons, was performed on 10 November 2022. In Table 1, the results revealed that both EWMA and the daily fresh blood comparison method consistently produced results within the permissible range of relative bias, adhering to external quality assessment standards. This consistency underscores the reliability of EWMA in capturing and correcting for potential biases in LDL-C measurements across different biochemical analyzers. In contrast, MM exhibited a higher bias in the comparison between the two reference biochemical instruments, particularly noticeable in the concentration range of 1.5–6.2 mmol/L on the fourth day. This deviation exceeded the allowable bias range, highlighting a potential limitation of MM in this specific context. The detailed findings are presented in Table 2 which encapsulates the comparison across various concentration ranges and judgment criteria. The observed higher bias in MM, particularly in specific concentration ranges, suggests a need for cautious interpretation when employing this method for inter-instrument comparisons in clinical settings. The consistent performance of EWMA across our analyses reinforces its suitability for monitoring analytical precision and ensuring QC in the measurement of LDL-C, a critical marker for CVD risk assessment.

Figure 2 illustrates the linear correlations that demonstrate the bias comparisons between two instruments.

Figure 3 provides a detailed scatter diagram analysis for the evaluation of fresh serum specimen measurements at critical medical decision levels.

## 4. Discussion

In the realm of contemporary clinical laboratory operations, the pursuit of result consistency across different instruments necessitates utilizing multiple strategies. These encompass the diligent execution of daily QC protocols, active engagement in regular external quality assessment programs, and the meticulous undertaking of periodic comparisons involving fresh blood samples. However, an inherent limitation of these approaches lies in their transient nature, which poses a noteworthy challenge in promptly detecting and effectively addressing systematic errors as they emerge [20]. To overcome these challenges, the integration of statistical rules, such as PBRTQC, can be highly beneficial [6]. PBRTQC involves the application of statistical methods to establish control limits based on the probability distribution of test results. Therefore, it is necessary to establish a more efficient, economical, and convenient program for real-time inter-instrument comparisons within the laboratory, which also has better detection abilities [5,7]. In this study, we investigated using the EP9A2 protocol, wherein specimens with varying concentration intervals were carefully selected for comparison. Our objective was to assess the performance of PBRTQC, specifically MM and EWMA. We evaluated these methods under standard detection conditions and scenarios.

The results of our study revealed a remarkable consistency between the EWMA method and fresh blood sample comparisons with standard detection. Notably, EWMA exhibited superior efficacy compared to MM. Our findings resonate with the study of Lu et al., who conducted a comprehensive assessment of real-time QC in clinical assays. Lu et al. also demonstrated that EWMA was adept at detecting very small quality-related changes, offering the added advantage of lower false-positive alarms compared to other methods [5]. These findings offer valuable insights into the potential application of EWMA as a supplementary tool for IQC. By incorporating EWMA into routine laboratory practices, it enables the continuous and effective monitoring of result consistency across different biochemical analyzers. This has significant implications for ensuring the reliability and accuracy of laboratory results. The utilization of EWMA as a complement to IQC represents a crucial advancement in laboratory practices. It also provides a means to promptly identify inconsistencies in results and take necessary corrective actions. This proactive approach is essential for maintaining the reliability of diagnostic and analytical processes, ultimately enhancing patient care, and contributing to the overall quality assurance in laboratory settings.

In such settings, where resources may be more constrained, EWMA offers a valuable approach for effectively monitoring result comparability. By selecting appropriate evaluation indices to assess the strengths and limitations of the model, the establishment of PBRTQC becomes achievable [8]. This study highlights the potential of EWMA in addressing the challenges associated with inter-device and inter-laboratory result comparisons. Its ability to mitigate patient population variability and enable real-time monitoring of result comparability signifies a significant advancement.

To date, there is a lack of reported research on the utilization of EWMA for inter-device or inter-laboratory comparisons of quantitative biochemical parameters. However, our study findings have provided evidence that the EWMA method exhibits reduced susceptibility to patient population variability. Moreover, our results have demonstrated that the comparability of results across different testing systems can be continuously and accurately monitored in real time, even when dealing with limited specimen volumes. These findings suggest that EWMA holds considerable promise for implementation in small- and medium-sized laboratories.

It has been over half a century since Hoffman and Wadi proposed the Patient Data Quality Control (PDQC) regulation in the 1960s [21]. The Committee on Analytical Quality of the International Federation of Clinical Chemistry and Laboratory Medicine (IFCLM) has also clearly stated that using PBRTQC in clinical practice should be encouraged [7,22]. It has shown substantial value and promise in clinical practice and research in laboratories of all sizes and has succeeded in becoming a next-generation quality management tool for the improvement of error detection [23,24]. Moreover, scientific studies have demonstrated the applicability of PBRTQC as an effective IQC method in the field of molecular diagnostics and point-of-care testing (POCT) [23,24].

These findings highlight the versatility and wide-ranging utility of PBRTQC across different aspects of healthcare [25]. While there is existing research on the application of PBRTQC, the studies and data available for its actual implementation remain limited [10]. To further enhance quality assurance and ensure accurate patient results, it is crucial to conduct additional analyses and investigations that explore the combination of PBRTQC with real-time inter-comparison across various clinical instrumentation types. By expanding the scope of research and implementing PBRTQC in real clinical settings, we can gain a more comprehensive understanding of its effectiveness and applicability. This broader use of PBRTQC can increase quality assurance measures and optimize patient outcomes. Therefore, we should prioritize additional studies and data collection focusing on the practical implementation of PBRTQC, particularly in the context of real-time inter-comparisons among different clinical instrumentation types [7]. These efforts will provide valuable insights and facilitate the broader adoption of PBRTQC within clinical settings, ultimately enhancing the accuracy and reliability of patients’ test results.

## 5. Conclusions

Our study demonstrates the practical implementation of combining PBRTQC with inter-instrument comparisons in the context of biochemical instrumentations. This integrated approach enables continuous patient data monitoring, thereby enhancing quality assurance within laboratory settings. A noteworthy finding from our research is the potential for PBRTQC to yield cost reductions and labor savings. By adopting PBRTQC, laboratories can optimize resource allocation and streamline operations, decreasing expenses and improving workforce efficiency.

## Figures and Tables

**Figure 1 diagnostics-14-00872-f001:**
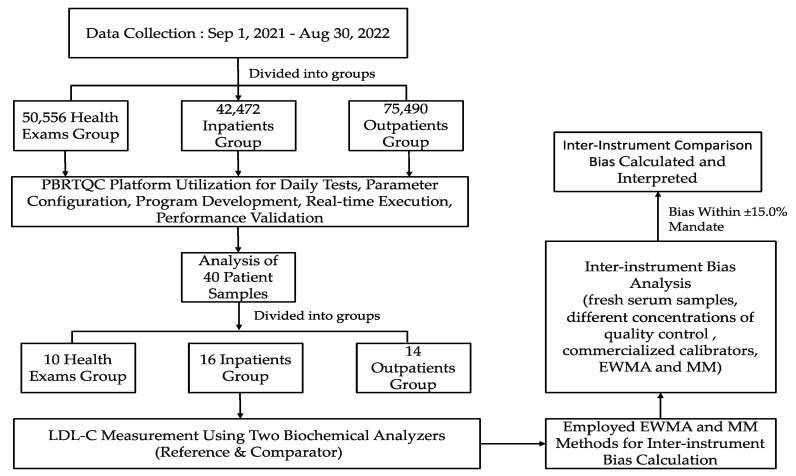
This illustration outlines the data collection process from a large patient database, the utilization of the PBRTQC intelligent monitoring platform for data analysis, the selection and analysis of 40 patient samples using two biochemical analyzers, and the calculation of inter-instrument bias employing EWMA and MM methods. Comparison of bias calculation methods and the impact of AI-based PBRTQC technology on enhancing laboratory efficiency, cost-effectiveness, and QC.

**Figure 2 diagnostics-14-00872-f002:**
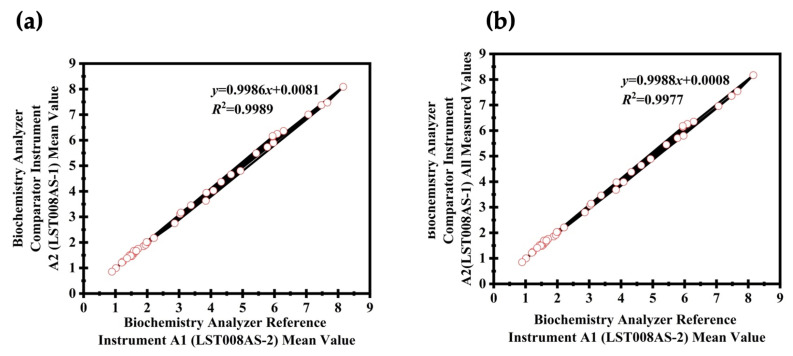
Linear correlations for bias comparisons of fresh serum specimens between two biochemistry analyzers. Graph (**a**) displays the linear correlation between the average values of two bio-chemistry analyzers, demonstrating a direct comparison of their measurement capacities. Graph (**b**) illustrates the linear correlation between the mean value of a reference biochemistry analyzer and the full spectrum of measurements from a comparator biochemistry analyzer, highlighting the consistency and accuracy of the comparator in comparison to the reference standard.

**Figure 3 diagnostics-14-00872-f003:**
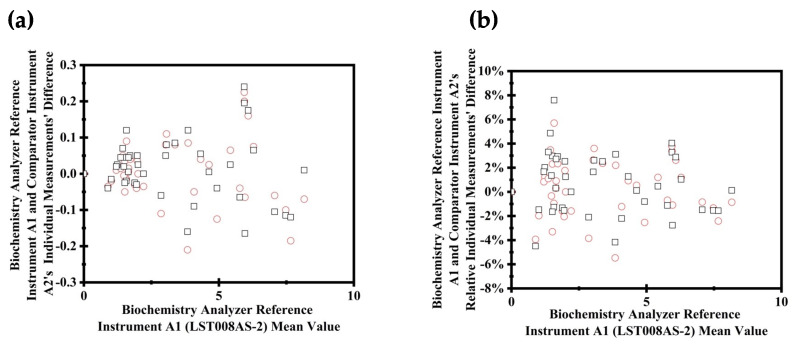
Scatter diagrams for comparing fresh serum specimen measurements at the medical decision levels. In both panels, squares represent the first round of measurements taken by the instruments, and circles indicate the second round of measurements. This figure shows the absolute and relative differences in measurements with reference instrument A1 and comparator instrument A2. Graph (**a**) illustrates the direct differences in individual measurements from the biochemistry analyzer reference instrument A1 and comparator instrument A2 versus the reference A1′s mean values, emphasizing the variability and bias between the instruments. Graph (**b**) focuses on the relative differences, as a percentage, providing insight into the consistency and precision of both instruments relative to reference instrument A1’s mean values. This comprehensive analysis aims to highlight the instruments’ quantitative and relative performance in evaluating fresh serum specimens.

**Table 1 diagnostics-14-00872-t001:** Fresh specimens’ comparison method for comparing bias results of LDL measurements across different biochemical instruments.

Test Name	Patient Group Concentration Range (mmoL/L)	Judgment Criteria ½ Total Error (15%)	Fresh Specimen Results							
			EP15A2 (Quality Target 15%)	EP9A2 (Quality Target 15%)						
				Medical Decision Level 1		Medical Decision Level 2		Medical Decision Level 3		Mean Bias % of Fresh Specimens
				3.4 mmoL/L		4.1 mmoL/L		4.9 mmoL/L		
			Mean Bias % of Fresh Specimens	Lower 95% Confidence Interval	Upper 95% Confidence Interval	Lower 95% Confidence Interval	Upper 95% Confidence Interval	Lower 95% Confidence Interval	Upper 95% Confidence Interval	
			EP15A2	EP9-A2	EP9-A2	EP9-A2	EP9-A2	EP9-A2	EP9-A2	EP9-A2
LDL (Total)	0.81–3.76	Bias	0.76%	−0.51%	0.83%	−0.49%	0.66%	−0.53%	0.57%	0.21%
		Conclusion	*	*	*	*	*	*	*	*
LDL (First Day)	1.2–6.4	Bias	0.47%	−5.09%	1.09%	−0.34%	1.11%	−0.22%	1.18%	−0.13%
		Conclusion	*	*	*	*	*	*	*	*
LDL (Second Day)	1.2–7.5	Bias	0.29%	−2.19%	−0.01%	−2.01%	−0.21%	−1.93%	−0.30%	−1.17%
		Conclusion	*	*	*	*	*	*	*	*
LDL (Third Day)	1.4–8.5	Bias	−4.30%	−3.84%	−1.34%	−3.37%	−1.39%	−3.08%	−1.34%	−2.50%
		Conclusion	*	*	*	*	*	*	*	*
LDL (Fourth Day)	1.5–6.2	Bias	−1.43%	2.37%	3.94%	2.33%	3.96%	2.24%	4.03%	3.22%
		Conclusion	*	*	*	*	*	*	*	*
LDL (Fifth Day)	1.4–6.2	Bias	1.19%	1.40%	2.93%	1.74%	3.10%	1.93%	3.31%	1.62%
		Conclusion	*	*	*	*	*	*	*	*

Explanation of symbols used in tables: symbol “*” denotes instances where the results or measurements are comparable.

**Table 2 diagnostics-14-00872-t002:** PBRTQC methods, different concentrations of QC, commercialized calibrators for comparing bias results of LDL measurements across different biochemical instruments.

Test Name	Patient Group Concentration Range (mmoL/L)	Judgment Criteria ½ Total Error (15%)	PBRTQC		QC Comparison Bias (15%)		Calibrator Comparison Bias (15%)
			MM (Quality Target 15%)	EWMA (Quality Target 15%)			
					Level 2	Level 3	Calibrator Bias%
			Bias %	Bias %	Average Bias %	Average Bias %	
			MM	EWMA	QC 2	QC 3	Calibrator
LDL (Total)	0.81–3.76	Bias	2.55%	0.62%	0.44%	0.07%	0.92%
		Conclusion	*	*	*	*	*
LDL (First Day)	1.2–6.4	Bias	1.26%	−2.04%	0.16%	−0.10%	0.33%
		Conclusion	*	*	*	*	*
LDL (Second Day)	1.2–7.5	Bias	2.03%	2.94%	0.00%	−0.98%	−1.14%
		Conclusion	*	*	*	*	*
LDL (Third Day)	1.4–8.5	Bias	14.01%	3.01%	−3.75%	−0.79%	−1.56%
		Conclusion	*	*	*	*	*
LDL (Fourth Day)	1.5–6.2	Bias	−24.66%	−1.08%	3.85%	1.91%	4.96%
		Conclusion	Incomparable	*	*	*	*
LDL (Fifth Day)	1.4–6.2	Bias	6.75%	1.25%	1.97%	0.30%	2.03%
		Conclusion	*	*	*	*	*

Explanation of symbols used in tables: symbol “*” denotes instances where the results or measurements are comparable. Note: the failure of MM comparison on day 4 was because there were only IP specimens and no OP specimens on Sunday. Moreover, 21 specimens on reference biochemical instrument (A1) and 13 specimens on reference biochemical instrument (A2) were originated from different clinical departments, so MM comparison did not pass.

## Data Availability

Data are contained within the article.

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
