# Peer review of "Integrating Patient-Based Real-Time Quality Control (PBRTQC) in a New Field: Inter-Comparison between Biochemical Instrumentations with LDL-C"

_diagnostics, 2024, doi:10.3390/diagnostics14090872_

Round 1

Reviewer 1 Report

Comments and Suggestions for Authors

Dear authors, 

Thank you very much for submitting this manuscript. The team addressed the effectiveness and application value of AI-based patient-based real-time quality control (PBRTQC) by comparing and evaluating LDL cholesterol test results to ensure real-time monitoring of the consistency between different biochemical instrumentations in clinical laboratories. The overall write-up is adequate, however there are rooms for improvements as the following:

1) The authors stated EWMA is lacking of data in addressing the reliability of the data, why the authors chose this method over others? Lack of literature supporting this in the introduction and the discussion as there are not enough data to compare with.

2) The retrospective data was collected in the duration of one year. What is the indication for choosing this data as only 160 samples were included? What happened to the rest? Please highlight this in the method.

3) Please improve Table 1 according to the MDPI format. I have difficulty in reading and internalizing the data.

4) How Table 1 and 2 compare each other? Can these two tables combined in one landscape table?

5) I think Figure 1 and 2 can be consolidated into a big figure. 

6) I think Figures 3 and 5 are insignificant. Just combine the comparison of Figures 4 and 6 with appropriate legends.

7) The discussion is adequate, however, there are lack of references to support the findings.

Reviewer 2 Report

Comments and Suggestions for Authors

The manuscript “Integrating Patient-based Real-time Quality Control (PBRTQC) In a New Field: Intercomparison between Biochemical Instrumentations with LDL - C” (diagnostics-2871138) by 𝐉𝐢𝐧𝐠𝐲𝐮𝐚𝐧 𝐖𝐚𝐧 and coauthors devoted to the evaluation of the “intercomparison between biochemical instrumentations for quality assurance” that is essential for hospital quality control (but in the local problems of the clinical laboratories concerning data on serum LDL-C). The authors wrote that “Patient based real-time quality control (PBRTQC) is a quality control method that monitors and evaluates the stability of the analytical performance of the testing process using real-time and continuous patient test results” and tried to prove these statements by some data on serum LDL-C. It is positive that the authors have used the following methods: floating average, normal average, median, and exponentially weighted moving average (EWMA) methods. The manuscript analyze the literature works in detail and at high level of discussion. I do not doubt the technical quality of the work and feel that there is a sufficient impact on a broader readership to justify publication in the "Diagnostics". The topic of this manuscript is in frame of the journal scopes, the subject matter is treated in depth. Thus, the present manuscript is actual and important, especially in the field of the development of clinical laboratories methods. But the similar approach is already published in the paper of the authors: Chao Song, Jun Zhou, Jun Xia, Deli Ye, Qian Chen, Weixing Li. Optimization and validation of patient-based real-time quality control procedure using moving average and average of normals with multi-rules for TT3, TT4, FT3, FT3, and TSH on three analyzers. J Clin Lab Anal. 2020;34:e23314. https://doi.org/10.1002/jcla.23314).

There are some comments:

1. In the “Abstract” (page 1, lines 17-18 Method:): the authors wrote “160 LDL cholesterol results were collected from outpatients and inpatients in the First Affiliated Hospital of Xi'an Jiaotong University”. It is not mentioned the amount of the patients, if these “outpatients and inpatients” are healthy or/and diseased patients ?! it is not clear: if the samples of the same patients are measured on both biochemical instruments or not ? This info must be mentioned in the “Abstract”.

2. Lines 106-107. There are no data on the amount of the patients in the “2.2. Materials and Methodology”. These data (including amount of the healthy and diseased patients, the age and sex of the “outpatients and inpatients”) must be included in the “2.2.1. Materials. Patient data collection” (i.e. below line 108). In addition, it is not clear: if the samples of the same patients are measured on both biochemical instruments or not ? This info is important for correct data presentation.

3. Lines 151-160 “Implementation of the comparison program”: the authors wrote “following the EP-09A2 protocol, we collected forty patient samples whose concentration classes were evenly distributed over the entire linear range of the method”. It is not clear: a) samples from what patients (healthy and/or diseased patients, the age and sex, the “outpatients and inpatients”) were taken ?; b) why only limited number of samples (“whose concentration classes were evenly distributed over the entire linear range of the method”), but not all samples were included in this study ?! It is no doubt that the amount of the patient samples can be increased significantly in the “First Affiliated Hospital of Xi'an Jiaotong University”, because the research have been done during the whole year..

4. Lines 151-160 “Implementation of the comparison program” : the authors wrote “Two rounds of testing were completed in five days of standard operation ….., with eight samples performed on each day…. The samples were performed either in the order of 1, 2, 3, 4, 5, 6, 7, 8 or 8, 7, 6, 5, 4, 3, 2, 1”. It is not clear why such experimental scheme have been chosen ?! It is necessary to provide the experimental scheme as separate Figure (compared to those as in the “FIGURE 1. Data transformation process in each analyzer and analyte in PBRTQC” in the paper of the authors: Chao Song, Jun Zhou, Jun Xia, Deli Ye, Qian Chen, Weixing Li. Optimization and validation of patient-based real-time quality control procedure using moving average and average of normals with multi-rules for TT3, TT4, FT3, FT3, and TSH on three analyzers. J Clin Lab Anal. 2020;34:e23314. https://doi.org/10.1002/jcla.23314).

5. It is well-known that the serum low density lipoprotein cholesterol (LDL-C) concentration is the dominant clinical parameter to judge a patient’s risk of developing cardiovascular disease (CVD). The discussion of this point is very important in the case of the data of diseased patients. If all samples are taken from the healthy patients, it is necessary to judge (to discuss), why only LDL-C clinical parameter are taken for this study. 

6. To my opinion, the authors can present the tables in more simple way and vertical format. For example, there are almost all data are comparable in both: Table 1. “fresh specimen comparison method for comparing the bias results of LDL comparison between different instruments” and Table 2. “PBRTQC method, different concentrations of quality control, commercialized calibrators for comparing the bias results of LDL comparison between different instruments”. That is why the word “comparable” can be exchange of the sing “*” with explanation in the appendix to these tables.  Moreover, some data not visible (i.e. missing) below the value “95%” in all lines No.5  in both Tables 1 and 2.

7. The presence of the “Abbreviation” part is valuable. For example, there are no explanation for the abbreviation “LDL-C” (low density lipoprotein cholesterol) in the text.

Comments on the Quality of English Language

Minor editing of English language required.

Reviewer 3 Report

Comments and Suggestions for Authors

Your study need some clarifications and I have made some comments and suggestions.

Comments follow throughout the attached document.

Round 2

Reviewer 2 Report

Comments and Suggestions for Authors

Dear authors,

I appreciate your constructive work on improving the manuscript “Integrating Patient-based Real-time Quality Control (PBRTQC) In a New Field: Intercomparison between Biochemical Instrumentations with LDL - C” (diagnostics-2871138). All my comments and suggestions were carefully considered and the appropriate corrections were made in the manuscript.

I propose to the editors to accept this manuscript in the present form.

Reviewer 3 Report

Comments and Suggestions for Authors

Please in line 67, identify the acronym and check the entire text again.

Author Response

RESPONSE SHEET #3

Manuscript ID: diagnostics-2871138

Dear Reviewer,

We would like to express our sincere gratitude to the dedication of your quality time to review our manuscript. We really appreciate your insightful feedback and constructive suggestions.
Please find attached thorough responses to each of your comments and concerns raised. We have carefully considered your feedback and made corresponding revisions. The corrections are highlighted and tracked in the resubmitted manuscript for your convenience and review.

Thank you once again for your invaluable contributions to refining our manuscript.

2. Questions for General Evaluation

Reviewer’s Evaluation

Response and Revisions

Does the introduction provide sufficient background and include all relevant references?

Yes

N/A

Are all the cited references relevant to the research?

Yes

N/A

Is the research design appropriate?

Yes

N/A

Are the methods adequately described?

Yes

N/A

Are the results clearly presented?

Yes

N/A

Are the conclusions supported by the results?

Yes

N/A

3. Point-by-point response to Comments and Suggestions for Authors

Comments 1: Please in line 67, identify the acronym and check the entire text again.

Response 1:

o    Changes in the article (marked in yellow): Page 2, line 57 – 58; Page 2, line 66-68; Page 2, line 65 – 66; Page 10, line 336 - 337.

Thank you for your valuable feedback. As per your request, I have identified and clarified the acronym mentioned in line 67. The acronym “quality control” stands for “QC.” To ensure clarity and comprehensibility throughout the manuscript, we have also reviewed the entire text to confirm that all acronyms are appropriately introduced and defined upon their first usage.

1.       “LDL-C has long been established as a critical marker for assessing cardiovascular disease (CVD) risk.” (Page 2, line 57 - 58)

2.       “Integrating PBRTQC into routine laboratory operations for LDL-C measurement not only facilitates the accurate assessment of CVD risk but also supports a patient-centered approach by incorporating real-time QC measures.” (Page 2, line 66 - 68)

3.       “Thus, Intercomparison between instrumentations for quality assurance is essential for laboratory QC” (Page 2, line 65 -66)

4.       “These findings offer valuable insights into the potential application of EWMA as a supplementary tool for IQC.” (Page 10, line 336 -337)

4. Response to Comments on the Quality of English Language

Point 1: I am not qualified to assess the quality of English in this paper.

Response 1: We understand the importance of ensuring that the manuscript meets the highest standards of clarity, fluency, and professionalism to communicate our findings to global audiences. We thoroughly reviewed and undertook additional revisions to address the language quality comprehensively. We also sought additional feedback from a native English-speaking colleague with expertise in academic writing to ensure that no aspect of language use detracts from the quality and comprehensibility of our research presented. Lastly, we are grateful for your constructive comment.

5. Additional clarifications

Response 1: N/A
